# Sequential Use of Sorafenib and Regorafenib in Hepatocellular Cancer Recurrence After Liver Transplantation: Treatment Strategies and Outcomes

**DOI:** 10.3390/cancers16223880

**Published:** 2024-11-20

**Authors:** Mehmet Fatih Ozbay, Hakan Harputluoglu, Mustafa Karaca, Omer Tekin, Mehmet Ali Nahit Şendur, Muhammed Ali Kaplan, Berksoy Sahin, Caglayan Geredeli, Fatih Teker, Deniz Tural, Sezer Saglam, Timuçin Çil, Ahmet Bilici, Cihan Erol, Ziya Kalkan, Ertugrul Bayram, Oguzhan Selvi, İlkay Gültürk, Sema Sezgin Göksu, Ali Murat Tatlı

**Affiliations:** 1Department of Medical Oncology, Kırsehir Training and Research Hospital, Kirsehir 40200, Turkey; mfozbay@hotmail.com; 2Department of Medical Oncology, Faculty of Medicine, Inonu University, Malatya 44000, Turkey; 3Department of Medical Oncology, Faculty of Medicine, Akdeniz University, Antalya 07070, Turkey; semasezgingoksu@gmail.com (S.S.G.); alimurattat@hotmail.com (A.M.T.); 4Department of Medical Oncology, Republic of Turkey Ministry of Health Ankara Bilkent City Hospital, Ankara 06800, Turkey; 5Department of Medical Oncology, Faculty of Medicine, Dicle University, Diyarbakır 21280, Turkey; 6Department of Medical Oncology, Faculty of Medicine, Çukurova University, Adana 01330, Turkey; 7Department of Medical Oncology, Health Sciences University Okmeydanı Training and Research Hospital, Istanbul 34098, Turkey; 8Department of Medical Oncology, Faculty of Medicine, Gaziantep University, Gaziantep 27410, Turkey; 9Department of Medical Oncology, Bakırköy Sadi Konuk Training and Research Hospital, Istanbul 34147, Turkey; 10Department of Medical Oncology, Demiroglu Bilim University Gayrettepe Florence Nightingale Hospital, Istanbul 34394, Turkey; 11Department of Medical Oncology, Adana City Training and Research Hospital, Adana Faculty of Medicine, University of Health Sciences, Adana 01230, Turkey; 12Department of Medical Oncology, Istanbul Medipol University, Istanbul 34815, Turkey

**Keywords:** hepatocellular carcinoma (HCC), liver transplantation, sorafenib, regorafenib, post-transplant recurrence

## Abstract

Liver cancer can sometimes come back after a liver transplant, which creates serious health challenges for patients. In this study, two medicines, sorafenib and regorafenib, were used to treat patients with this type of recurring cancer. Patients started with sorafenib, and if this medicine stopped working or caused too many side effects, they switched to regorafenib. The results showed that using these medicines one after the other could help patients live longer. However, these treatments can also lead to strong side effects, so patients need careful monitoring. This study highlights the importance of personalized care for people facing a return of liver cancer after a transplant.

## 1. Introduction

Hepatocellular carcinoma (HCC) is recognized as a major public health problem worldwide and is particularly prevalent among individuals with chronic liver disease [1,2].The most common risk factors for HCC include chronic hepatitis B and C infections, alcohol use, and non-alcoholic fatty liver disease (NAFLD). The incidence of HCC is higher in Asian and African regions where hepatitis B and C are endemic but has also been increasing in Western countries in recent years, which is associated with an increase in NAFLD and alcohol consumption [3,4,5,6].

Curative treatment options such as surgical resection, ablation therapies, and liver transplantation are available for HCC patients diagnosed at early stages [7,8]. However, since the majority of patients are diagnosed at an advanced stage, these curative options are not applicable, and systemic therapies come to the fore for these patients. Liver transplantation is considered to be one of the most effective methods in the treatment of HCC, especially for suitable patients in the early stage. However, the risk of recurrence of HCC after transplantation is a critical factor that negatively affects the long-term survival of patients. In the literature, HCC recurrence rates after liver transplantation vary between 10% and 20%, which poses a significant challenge in clinical management [9,10,11]. 

Treatment options for HCC patients who relapse after liver transplantation are very limited, and this patient group is usually excluded from clinical trials. The management of HCC recurrence after liver transplantation is particularly complex, as these patients are already on immunosuppressive therapy, which can impact the efficacy of various treatment modalities. Existing evidence suggests that the tyrosine kinase inhibitors sorafenib and regorafenib may have a role in the treatment of post-transplant HCC recurrence [12,13,14]. 

Sorafenib has been the standard of care for advanced, unresectable HCC for over a decade, demonstrating improved overall survival compared to placebo. However, the use of sorafenib in the post-transplant setting is complicated by potential drug–drug interactions with immunosuppressive agents and higher rates of treatment-related toxicity [15,16]. After progression on sorafenib, the multikinase inhibitor regorafenib has shown efficacy as a second-line treatment option in HCC, including in the post-transplant setting [17,18,19]. Recent research has also explored the potential of immunotherapy agents, such as nivolumab, in HCC patients who have failed prior sorafenib treatment. However, the use of immunotherapy in the post-transplant setting remains limited due to concerns about graft rejection and insufficient clinical data [20,21,22].

Understanding the unique challenges and treatment strategies for HCC recurrence after liver transplantation is crucial, as this patient population is often excluded from large clinical trials. The sequential use of sorafenib and regorafenib in this setting may offer a viable treatment approach, but further research is needed to optimize outcomes and management of treatment-related toxicities in this complex patient group.

## 2. Materials and Methods

This study is a retrospective, multicenter analysis aimed at evaluating the effectiveness of sequential sorafenib and regorafenib treatments in patients with HCC recurrence following liver transplantation. The study was conducted on 73 patients diagnosed with recurrent HCC between 2012 and 2022. Patient data were collected from 11 oncology centers across 7 different cities in Turkey.

The study population included patients aged 18 years or older with a confirmed diagnosis of HCC recurrence following liver transplantation. Eligible patients began treatment with sorafenib, followed by regorafenib upon progression or intolerance to sorafenib. Patients were excluded if they had incomplete medical records or had received systemic therapies other than sorafenib or regorafenib.

The study population consisted of patients aged 18 years or older with HCC who had undergone liver transplantation and experienced HCC recurrence after the transplant. Eligible patients had received sorafenib as the first-line treatment following recurrence and were subsequently treated with regorafenib either due to intolerance to sorafenib or disease progression. Exclusion criteria included patients who had undergone multiple organ transplants, those with HCC recurrence within 3 months of liver transplantation, patients with severe organ failure (such as heart or renal failure) unrelated to HCC, and those who were participating in other clinical trials.

### 2.1. Data Collection

Clinical and demographic data, including patient age, sex, body mass index (BMI), date of liver transplantation, underlying liver disease, and cirrhosis etiology, were collected from medical records. Additionally, data on tumor characteristics, such as the size and number of recurrent lesions, as well as the time to recurrence after transplantation, were obtained. The performance status of patients was assessed using the Eastern Cooperative Oncology Group (ECOG) score, and laboratory data, including alpha-fetoprotein (AFP) levels and liver function tests (ALT, AST, bilirubin), was recorded. Liver function status was evaluated using the Child–Pugh score. Information on post-transplant immunosuppressive therapy regimens was also gathered. Treatment-related data included the dosage and duration of sorafenib and regorafenib, the reasons for treatment discontinuation (e.g., progression or intolerance), and any reported adverse effects, along with management strategies.

### 2.2. Treatment Protocol

All patients received sorafenib as first-line therapy upon the detection of HCC recurrence following liver transplantation. Patients who experienced intolerance to sorafenib, defined as the occurrence of grade 3 or higher toxicities unmanageable with dose adjustments or disease progression, as determined by radiological evidence of tumor growth according to RECIST version 1.1 criteria, were transitioned to second-line treatment with Regorafenib. Radiological evaluations were performed every 8 to 12 weeks to monitor treatment response and disease progression. The dose and duration of each therapy were recorded, along with any necessary dose modifications.

### 2.3. Ethical Approval 

The study was designed and conducted in accordance with internationally recognized principles of Good Clinical Practice and the Declaration of Helsinki. Ethical approval for the study was obtained from the Akdeniz University Clinical Research Ethics Committee.(Protocol #2012-KAEK-20) with decision number KAEK-726, granted on 13 October 2021.

### 2.4. Outcome Measures

The primary endpoints were progression-free survival (PFS) for both sorafenib (PFS1) and regorafenib (PFS2), as well as overall survival (OS). PFS was defined as the time from the start of each treatment to disease progression or death from any cause. OS was measured from the date of recurrence diagnosis until death or last follow-up. The secondary endpoints included the incidence and severity of adverse effects, categorized according to the Common Terminology Criteria for Adverse Events (CTCAE).

### 2.5. Statistical Analysis

Survival analyses were performed using the Kaplan–Meier method to estimate PFS and OS. The Cox proportional hazards model was used to assess risk factors associated with progression and mortality, with hazard ratios (HR) and 95% confidence intervals (CI) calculated. Variables included in the analysis were ECOG performance status, Child–Pugh classification, AFP levels, and metastasis location. The analyses were performed with the SPSS 26.0 program, and a 95% confidence level was used. In the analyses, the relationship between categorical variables was analyzed using the Chi-square test. The measurements in terms of categorical variables were analyzed using a *t*-test. The relationship between the measurements was analyzed using Pearson’s correlation test. Survival for PFS1, PFS2, and OS was analyzed using the Kaplan–Meier test.

## 3. Results

### 3.1. Patient Demographics and Clinical Characteristics

The 73 patients included in the study were diagnosed with recurrent HCC after liver transplantation. Of these patients, 84.9% were male (*n* = 62) and 15,1% were female (*n* = 11). The mean age of the patients was 56.5 ± 11.4 years, and the mean age at diagnosis was 52.3 ± 11 years. The AFP levels of the patients at the time of diagnosis were widely distributed, and the mean AFP level was 1790.1 ± 8974.0 ng/mL. Furthermore, cirrhosis was detected in 75.3% of the patients, and hepatitis B virus (HBV) was the most common etiology of cirrhosis among these patients (72.6%).

The patients’ post-transplant period until relapse varied between (5–83.17 months), with a median of 13.72 months. The mean duration was found to be 20.19 ± 19.06 months.

All patients received sorafenib as first-line treatment. Among patients who experienced progression with sorafenib or discontinued treatment due to toxicity, 45.2% (*n* = 33) continued treatment with regorafenib. The mean follow-up period after recurrent disease was 39.65 months (Table 1).

### 3.2. Sorafenib Treatment

The performance status of sorafenib-treated patients was categorized as ECOG 0 in 32.9%, ECOG 1 in 61.6%, and ECOG 2 in 5.5%. The mean number of cures achieved during treatment was 8.6 ± 8.3 (median: 6). While 86.5% of patients had to discontinue treatment due to progression, 6.8% discontinued treatment due to toxicity. Clinical response rates with sorafenib were complete response (CR) 2.8%, partial response (PR) 19.7%, stable disease (SD) 33.8%, and progressive disease (PD) 43.7% (Table 2).

### 3.3. Progression-Free Survival 1 (PFS1) with Sorafenib

The median progression-free survival (PFS) calculated after sorafenib treatment was 5.6 months (SE: 0.3). Three-month PFS rate was 67.8% (SE: 5.5), six-month rate 39.9% (SE: 5.9), one-year rate 24.3% (SE: 5.3), two-year rate 12.2% (SE: 4.2) and three-year rate 3.0% (SE: 2.7). These results suggest that sorafenib treatment may be effective for a certain period of time in HCC patients, but long-term survival remains limited (Table 3, Figure 1).

### 3.4. Regorafenib Treatment

The performance status of the patients who received regorafenib after sorafenib treatment was evaluated as ECOG 0 in 21.2%, ECOG 1 in 60.6%, and ECOG 2 in 18.2%. The median duration of treatment was 7.6 cycles. The starting dose of regorafenib was generally 120 mg (45.5%) and 160 mg (24.2%), and 69.7% of patients received dose escalation during treatment. In terms of treatment responses, the partial response (PR) rate was 51.5%, and the progressive disease (PD) rate was 48.5%. Treatment was discontinued in 87.9% of the patients due to progression and 6.1% due to toxicity (Table 4).

### 3.5. Progression Free Survival (PFS) with Regorafenib

The median progression-free survival time after regorafenib treatment was calculated as 5.9 months (SE: 1.0). Three-month survival rate was 62.6% (SE: 8.6), six-month 38.2% (SE: 8.9), one-year 27.8% (SE: 8.3), two-year 17.4% (SE: 7.0) and three-year 3.5% (SE: 3.4) (Table 5, Figure 2).

### 3.6. Overall Survival (OS)

The median overall survival (OS) calculated throughout the study was 35.9 months (SE: 6.8). Overall survival rates were 93.2% (SE: 2.9) at one year, 55.0% (SE: 5.8) at three years, 36.0% (SE: 5.7) at five years and 12.1% (SE: 4.4) at 10 years (Figure 3, Table 6).

### 3.7. Response to Treatment According to Child–Pugh Classification

Cox regression analysis examined baseline factors affecting PFS, including Child–Pugh score, ECOG status, and AFP levels. Child–Pugh A patients showed longer PFS and OS (14 and 22 months) compared to Child–Pugh B (10 and 16 months), but the difference was not statistically significant (HR: 1.364, 95% CI: 0.492–3.783, *p* = 0.551).

ECOG status significantly impacted PFS, with ECOG 2 patients at higher risk of progression (HR: 3.749, 95% CI: 1.071–13.123, *p* = 0.039). AFP levels and adverse reactions, such as hand and foot syndrome and fatigue, had no significant effect on PFS.

During sorafenib treatment, 78.4% of patients experienced adverse reactions, with hand and foot syndrome (41.9%) and fatigue (62.2%) being the most common. Similarly, regorafenib treatment was associated with adverse reactions in 81.8% of patients, predominantly fatigue (78.8%) and hypertension (33.3%).

## 4. Discussion

This study comprehensively evaluated the efficacy and survival effects of sorafenib and Regorafenib in patients with HCC relapsed after liver transplantation. The findings indicate that PFS and OS times can be significantly improved in this patient group, but some remarkable differences and similarities also emerge when these findings are compared with the literature.

In our study, the median PFS duration (PFS1) after Sorafenib treatment was found to be 5.6 months. This finding is consistent with the PFS duration of sorafenib reported in previous studies [23,24]. For example, in the SHARP study, the PFS duration of sorafenib in patients with advanced HCC was reported as 5.5 months. This supports the idea that sorafenib may be effective in delaying disease progression in patients with advanced HCC [25]. However, prognostic factors such as ECOG performance status and AFP levels were observed to have significant effects on PFS1. In the literature, it is frequently emphasized that these factors are critical factors affecting treatment response and survival times [26,27,28]. 

Regorafenib treatment was used as second-line therapy in patients who developed resistance to sorafenib, and the median PFS duration (PFS2) was 5.9 months. This finding indicates a longer duration compared to the PFS of 3.1 months reported in the RESORCE study [29]. This difference may be explained by factors such as differences in the patient population and variations in the duration of follow-up. Regorafenib is known to be particularly beneficial in patients who show resistance to sorafenib treatment, indicating that it may be an important option as a second-line treatment in this patient group [30,31,32]. 

Our study’s patient cohort, with an average age of 61.5 years and a predominance of male patients (84.9%), mirrors the common demographic characteristics seen in hepatocellular carcinoma (HCC) cases following liver transplantation. This aligns with findings in prior research, such as Kwon et al., who observed that 89.5% of patients treated with either sorafenib or regorafenib after transplant were male, and Ren et al., whose study of advanced HCC patients undergoing sequential therapy reported a similar gender distribution (90.7% male). Additionally, Iavarone’s study further supports these trends, linking male prevalence in HCC to specific risk factors such as lifestyle and hepatitis infections. While Kwon et al. reported comparable demographics, their study’s smaller patient groups limited the ability to demonstrate a survival benefit across treatment cohorts [33,34]. 

In the study conducted by Iavarone et al., the median progression-free survival (PFS1) for sorafenib in post-liver transplant patients with recurrent hepatocellular carcinoma (HCC) was 3.0 months, reflecting the limited efficacy of sorafenib in this specific patient population. In contrast, our study reported a slightly longer median PFS1 of 5.6 months for sorafenib, which might be attributed to differences in patient characteristics or regional treatment protocols. Similarly, for regorafenib, Iavarone et al. observed a median PFS of 5.5 months, which closely aligns with the median PFS2 of 5.9 months reported in our study. When examining overall survival (OS), Iavarone et al. reported a total median OS of 28.8 months for patients treated with sorafenib followed by regorafenib. In comparison, our study observed a longer median OS of 35.9 months. The difference in OS may be influenced by variations in patient cohorts, follow-up durations, and treatment strategies, particularly considering the potential for improved liver function in post-transplant patients. Both studies highlight that the sequential use of sorafenib and regorafenib offers substantial survival benefits [35]. 

In our analysis, we observed a median OS of 35.9 months with the sequential use of both treatments. This is longer than what has typically been reported in the literature, suggesting that there might be additional factors contributing to the better outcomes seen in our cohort. Notably, a significant proportion of our patients (75%, as shown in Table 1) were diagnosed with cirrhosis at the time of their HCC diagnosis. The removal of the cirrhotic liver during transplantation likely played a key role in extending survival for these patients. As Mazzaferro et al. highlighted, liver transplantation not only treats the cancer itself but also eliminates the cirrhotic burden, which can have a meaningful impact on survival [36]. This aligns well with our findings and supports the idea that reducing cirrhosis through transplantation may contribute to improved outcomes, regardless of anti-cancer therapies. This unique aspect of our cohort likely explains the observed survival advantage and underscores the importance of accounting for cirrhosis status in post-transplant evaluations [37,38,39].

In terms of side effects, side effects such as hand and foot syndrome, fatigue, hypertension, and diarrhea were frequently observed in both sorafenib and regorafenib treatments. These findings are in line with the side effect profiles reported in previous studies. The management of side effects is critical during the treatment process, and careful monitoring of potential complications that may lead to disruption of the treatment process is required in these patients. In particular, a higher incidence of side effects in sorafenib-treated patients may adversely affect treatment continuity and thus shorten survival times. In regorafenib treatment, the severity and frequency of side effects are more variable but considering that this drug is used in patients who have previously developed resistance to treatment, the management of side effects gains greater importance [40,41].

This study has several limitations that should be acknowledged. First, the retrospective design may introduce inherent biases related to data collection and patient selection. The lack of a control group restricts our ability to draw definitive conclusions regarding the survival benefit of sequential sorafenib and regorafenib therapy. Additionally, our relatively small sample size, particularly in patients transitioning from sorafenib to regorafenib, limits the generalizability of our findings. Lastly, the absence of detailed molecular and biomarker analyses restricts a more nuanced understanding of individual treatment responses, highlighting the need for future studies to incorporate these elements.

## 5. Conclusions

In conclusion, our findings suggest that sequential sorafenib and regorafenib therapy may provide a survival benefit for patients with recurrent HCC following liver transplantation. Despite the limitations of a retrospective design and lack of a control group, our study contributes valuable insights into the potential efficacy of this therapeutic approach. Further prospective studies with larger cohorts and control groups are needed to validate these results and optimize treatment strategies. Understanding patient-specific factors, including baseline liver function and performance status, may aid in tailoring therapies for improved outcomes in this unique patient population.

## Figures and Tables

**Figure 1 cancers-16-03880-f001:**
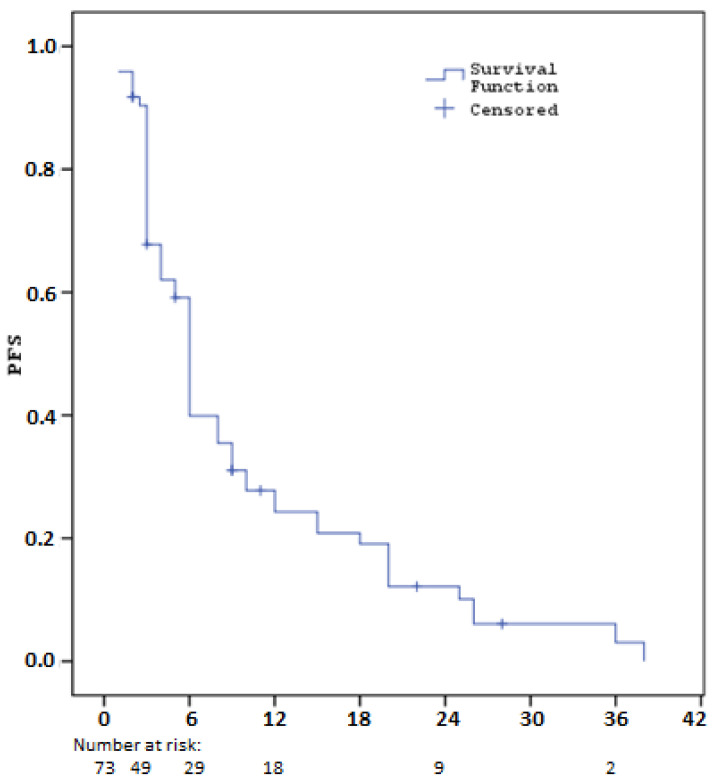
Kaplan–Meier curves for progression-free survival with Sorafenib (PFS1).

**Figure 2 cancers-16-03880-f002:**
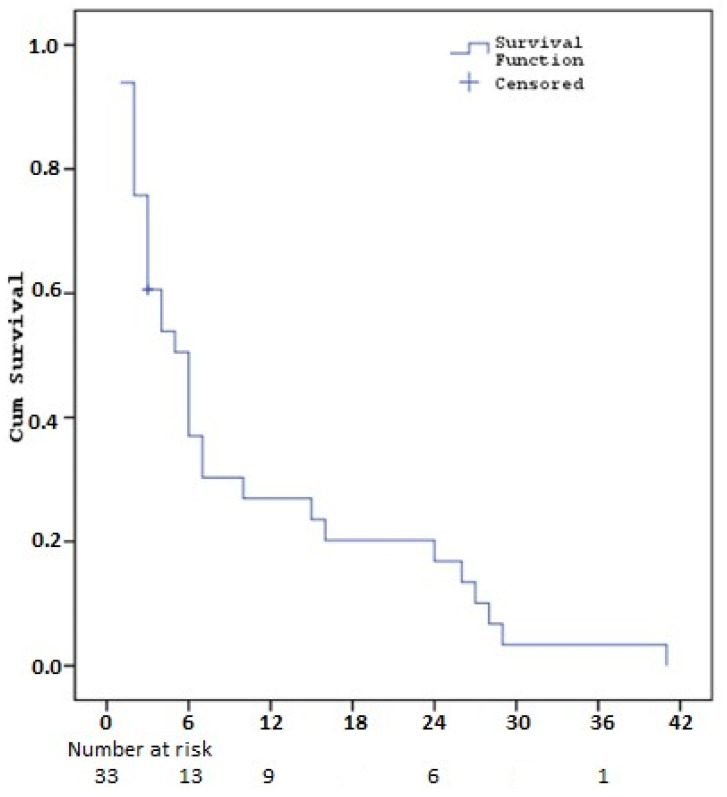
Kaplan–Meier curves for progression-free survival with regorafenib (PFS2).

**Figure 3 cancers-16-03880-f003:**
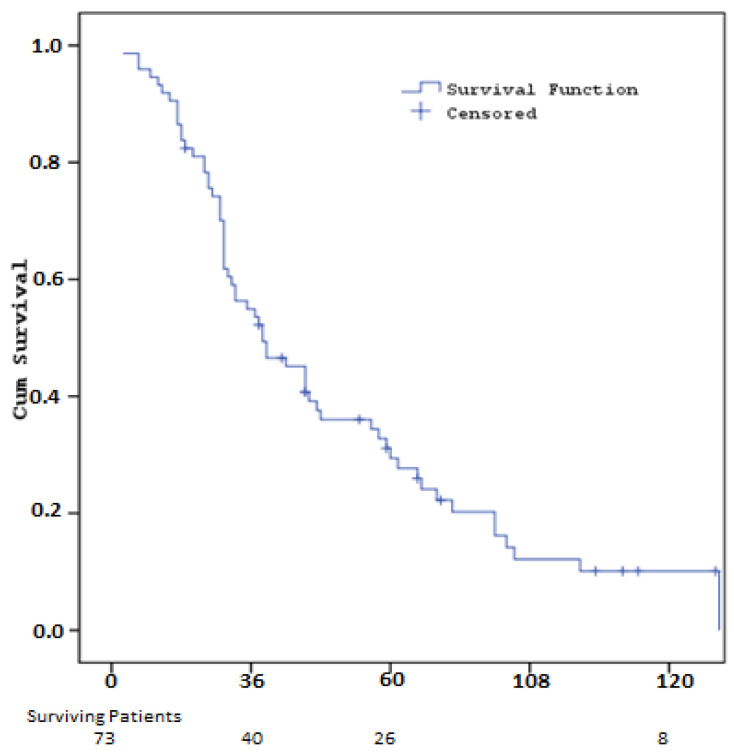
Kaplan–Meier curves for Overall Survival.

**Table 1 cancers-16-03880-t001:** Demographic and clinical characteristics of patients with recurrent HCC after liver transplantation.

Parameter	Value
Gender *n* (%)	Female: 11 (15.1%), Male: 62 (84.9%)
Age (Mean ± SD) Min–Max (Median)	56.5 ± 11.4 (22–74) (59)
Age at Diagnosis (Mean ± SD) Min–Max (Median)	52.3 ± 11.0 (19–68) (54)
AFP at Diagnosis (Mean ± SD) Min–Max (Median)	1790.1 ± 8974 (1–73,593) (566)
ECOG at Diagnosis *n* (%)	0: 29 (39.2%), 1: 44 (59.5%), 2: 1 (1.4%)
Cirrhosis at Diagnosis *n *(%)	No: 18 (24.7%), Yes: 55 (75.3%)
Etiology of Cirrhosis *n* (%)	HBV: 53 (72.6%), HCV: 7 (9.6%), Alcohol: 2 (2.7%), Other: 11 (15.1%)
Single Lesion *n* (%)	Less than 5 cm: 42 (59.2%), Greater than 5 cm: 29 (40.8%)

**Table 2 cancers-16-03880-t002:** Summary of sorafenib treatment parameters and outcomes in patients with recurrent HCC after liver transplantation.

Parameter	Value
AFP Before Sorafenib (Mean ± SD) Min–Max (Median)	2151 ± 8632 (0.77–54,000) (100)
Child Score Before Sorafenib *n* (%)	A: 69 (94.5%), B: 4 (5.5%)
ECOG Before Sorafenib *n* (%)	0: 24 (32.9%), 1: 45 (61.6%), 2: 4 (5.5%)
Number of Sorafenib Cycles (Mean ± SD) Min–Max (Median)	8.6 ± 8.3 (1–38) (6)
Discontinuation of Sorafenib *n* (%)	Progression: 64 (86.5%), Toxicity: 5 (6.8%), Continuing: 5 (6.8%)
Best Response to Sorafenib *n* (%)	CR: 2 (2.8%), SD: 24 (33.8%), PR: 14 (19.7%), PD: 31 (43.7%)
Adverse Reactions *n* (%)	No: 16 (21.6%), Yes: 58 (78.4%)
Hand and Foot Syndrome *n* (%)	31 (41.9%)
Hand and Foot Syndrome Grade *n* (%)	Grade 1: 15 (20.3%), Grade 2: 12 (16.2%), Grade 3: 2 (2.7%), Grade 4: 1 (1.4%)
Fatigue *n* (%)	46 (62.2%)
Fatigue Grade *n* (%)	Grade 1: 16 (21.6%), Grade 2: 20 (24.0%), Grade 3: 10 (13.5%)
Hypertension *n* (%)	18 (24.3%)
Hypertension Grade *n* (%)	Grade 1: 6 (8.1%), Grade 2: 12 (16.2%)
Diarrhea *n* (%)	27 (36.5%)
Diarrhea Grade *n* (%)	Grade 1: 10 (13.5%), Grade 2: 11 (14.9%), Grade 3: 5 (6.8%)
Rash *n* (%)	8 (10.8%)
Rash Grade *n* (%)	Grade 1: 3 (4.1%), Grade 2: 3 (4.1%), Grade 4: 2 (2.7%)

**Table 3 cancers-16-03880-t003:** PFS values for sorafenib treatment by time.

Parameter	Value
PFS Duration (Months) Median (SE)/95% CI	5.6 (SE: 0.3)/5.4–6.6
PFS (%) at 3 months	67.8% (SE: 5.5%)
PFS (%) at 6 months	39.9% (SE: 5.9%)
PFS (%) at 1 year	24.3% (SE: 5.3%)
PFS (%) at 2 years	12.2% (SE: 4.2%)
PFS (%) at 3 years	3.0% (SE: 2.7%)

**Table 4 cancers-16-03880-t004:** Summary of regorafenib treatment parameters and outcomes.

Parameter	Value
ECOG Before Regorafenib *n* (%)	0: 7 (21.2%), 1: 20 (60.6%), 2: 6 (18.2%)
Child Score Before Regorafenib *n* (%)	A: 31 (93.9%), B: 2 (6.1%)
Number of Regorafenib Cycles (Mean ± SD) Min–Max (Median)	7.6 ± 8.2 (2–36) (4)
Previous Treatment Lines Before Regorafenib *n* (%)	1: 23 (71.9%), 2: 7 (21.9%), 3: 2 (6.3%)
Previous Treatments Before Regorafenib *n* (%)	Sorafenib: 27 (81.8%), TACE: 1 (3.0%) *, Sorafenib + TACE: 3 (9.1%) **, Sorafenib + TARE: 1 ** (3.0%), Sorafenib + TACE + TARE: 1 (3.0%) **
Initial Dose *n* (%)	80 mg: 10 (30.3%), 120 mg: 15 (45.5%), 160 mg: 8 (24.2%)
Dose Increase During Follow-up *n* (%)	No: 10 (30.3%), Yes: 23 (69.7%)
Maintenance Dose *n* (%)	120 mg: 12 (36.4%), 160 mg: 21 (63.6%)
Dose Reduction *n* (%)	No: 24 (72.7%), Yes: 9 (27.3%)
Treatment Discontinued *n* (%)	No: 11 (33.3%), Yes: 22 (66.7%)
Reason for Discontinuation *n* (%)	Progression: 29 (87.9%), Toxicity: 2 (6.1%), Continuing: 2 (6.1%)
Progression Location *n* (%)	Liver: 10 (30.3%), Lung: 10 (30.3%), Bone: 9 (27.3%), Brain: 0 (0.0%), Abdomen: 12 (36.4%)
Best Response *n* (%)	PR: 17 (51.5%), PD: 16 (48.5%)
Marker Response *n* (%)	No: 18 (56.3%), Yes: 14 (43.8%)
Adverse Reactions *n* (%)	No: 6 (18.2%), Yes: 27 (81.8%)
Fatigue *n* (%)	26 (78.8%)
Fatigue Grade *n* (%)	Grade 1: 8 (24.2%), Grade 2: 15 (45.5%), Grade 3: 3 (9.1%)
Hypertension *n* (%)	11 (33.3%)
Hypertension Grade *n* (%)	Grade 1: 5 (15.2%), Grade 2: 5 (15.2%), Grade 3: 1 (3.0%)
Diarrhea *n* (%)	10 (34.5%)
Diarrhea Grade *n* (%)	Grade 1: 6 (18.2%), Grade 2: 4 (12.1%)
Rash *n* (%)	8 (24.2%)
Rash Grade *n* (%)	Grade 1: 4 (12.1%), Grade 2: 3 (9.1%), Grade 3: 1 (3.0%)

ECOG: Eastern Cooperative Oncology Group performance status, TACE: Transarterial Chemoembolization, TARE: Transarterial Radioembolization PR: Partial Response, PD: Progressive Disease. * Patient 41 received only TACE initially after liver transplantation due to a localized hepatic lesion. Systemic recurrence was later observed, leading to the initiation of sorafenib treatment. ** Received both treatments concurrently as part of their first-line management.

**Table 5 cancers-16-03880-t005:** PFS values for regorafenib treatment by time.

Parameter	Value
PFS Duration (Months) Median (SE)/95% CI	5.9 (SE: 1.0)/4–8
PFS (%) at 3 months	62.6% (SE: 8.6%)
PFS (%) at 6 months	38.2% (SE: 8.9%)
PFS (%) at 1 year	27.8% (SE: 8.3%)
PFS (%) at 2 years	17.4% (SE: 7.0%)
PFS (%) at 3 years	3.5% (SE: 3.4%)

**Table 6 cancers-16-03880-t006:** Overall Survival (OS) durations and percentages at various time points.

Parameter	Value
OS Duration (Months) Median (SE)/95% CI	35.9 (SE: 6.8)/25.7–52.3
OS (%) at 1 year	93.2% (SE: 2.9%)
OS (%) at 3 years	55.0% (SE: 5.8%)
OS (%) at 5 years	36.0% (SE: 5.7%)
OS (%) at 10 years	12.1% (SE: 4.4%)

## Data Availability

The data presented in this study are available on request from the corresponding author due to privacy and ethical restrictions.

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
