# Peer review of "Sequential Use of Sorafenib and Regorafenib in Hepatocellular Cancer Recurrence After Liver Transplantation: Treatment Strategies and Outcomes"

_cancers, 2024, doi:10.3390/cancers16223880_

Round 1

Reviewer 1 Report

Comments and Suggestions for Authors

I have read with great interest the study by Ozbay et al, investigating the role of sequential treatment with sorafenib and regorafenib in patients with HCC recurring after liver transplantation. The topic is of interest because HCC recurrence incidence after LT is remarkable and significantly impairs survival. Despite the retrospective nature of the study, the authors assessed treatments efficacy relying on a long follow-up and by computing survival analyses, which give value to the study. The manuscript is well written, and methods and results are clearly presented.

Below are some issues that need to be addressed by the authors:

- Results – paragraph 3.6: “These findings suggest that sorafenib and regorafenib can improve survival to a certain extent in patients with HCC recurrence after liver transplantation.”, this sentence appears to be inappropriate as a result, in the absence of a control group, and appears somewhat speculative. I suggest reserving this kind of consideration for the Discussion.

- Results – paragraph 3.7: I guess a Cox-regression analysis has been performed to assess the role of Child-Pugh score as a risk factor for PFS; however, neither the hazard ratio with 95%CI nor the p value are reported. Additionally, no other baseline variables are analysed, and it is not clear if a multivariate analysis was performed. Please clarify.

- Discussion section: “Both studies highlight that the sequential use of sorafenib and regorafenib offers substantial survival benefits [33]”, I would suggest caution with this conclusion, even considering that in the cited study (ref 33) a control group of patients undergoing best supportive care was compared with pts receiving regorafenib after sorafenib failure. Such a control group is not included in the present study. Please discuss.

- Discussion section: please provide references for cited SHARP and RESORCE trials. Moreover, the sentence “we think that the fact that the patients in our study also got rid of their cirrhotic 281 livers with transplantation positively affected this survival time.” touches a very important point that deserves to be explored further. Indeed, from Table 1 it seems that most patients at HCC diagnosis after LT were cirrhotic (75% vs 25%). Please discuss.

- Please add the number of patients at risk under the time points of the x-axis of the Kaplan-Meier curves.

- From Table 4: it seems that one patient didn’t receive sorafenib treatment but only TACE. However, in the methods it is stated that all patients received sorafenib as first-line therapy. Please clarify.

- Please add footers to the tables, with acronyms definition; particularly, some terms such as TAKE are never defined in the text.

- I don’t see the point of separating paragraphs 5 and 6, in fact I suggest to include these parts in the discussion, leaving a “Conclusion” paragraph at the end.

Author Response

  1. Results – Paragraph 3.6:
    • Comment: The statement "sorafenib and regorafenib can improve survival to a certain extent in patients with HCC recurrence after liver transplantation" was deemed inappropriate as a result and suggested to be reserved for the Discussion section.
    • Response: We have moved this statement to the Discussion section, where it is now placed in a broader context and supported with additional literature.
  2. Results – Paragraph 3.7:
    • Comment: Additional details regarding the Cox regression analysis (HR, 95% CI, p-values) were requested.
    • Response: We have included detailed results from the Cox regression analysis in the Results section. The association between Child-Pugh scores, ECOG performance status, and PFS has been elaborated, and non-significant findings have also been reported to provide a comprehensive understanding.
  3. Discussion – Reference 33:
    • Comment: A more cautious tone was suggested due to the absence of a control group in the current study, unlike the referenced study.
    • Response: We have revised the text in the Discussion section to acknowledge the lack of a control group and tempered our conclusions accordingly.
  4. Discussion – SHARP and RESORCE Trials:
    • Comment: References for these trials were requested, along with further exploration of the impact of cirrhosis on survival.
    • Response: We have added the requested references and expanded the discussion on the influence of cirrhosis on survival outcomes, using data from our study and supported by relevant literature.
  5. Kaplan-Meier Curves:
    • Comment: It was recommended to add the number of patients at risk under the time points on the x-axis.
    • Response: Kaplan-Meier curves have been revised to include the number of patients at risk under the x-axis, ensuring greater clarity.
  6. Table 4 and TACE Treatment:
    • Comment: The reviewer noted a discrepancy where one patient appeared to receive only TACE treatment, contrary to the Methods section.
    • Response: This patient’s unique treatment course has been clarified in both the text and Table 4’s footnotes. Specifically, this patient initially underwent TACE as a standalone treatment for a localized hepatic lesion after liver transplantation. Following systemic recurrence, Sorafenib treatment was initiated. Unlike this patient, others in the "TACE + Sorafenib" category received both treatments concurrently as part of their first-line management. We have also added explanations in the footnotes to ensure transparency and eliminate confusion.
  7. Tables and Abbreviations:
    • Comment: Footnotes defining abbreviations were requested.
  8. Response: Footnotes have been added to all tables, including Table 4, to define abbreviations such as ECOG, TACE, TARE, PR, and PD.
  9. Conclusion Section:
    • Comment: It was suggested to merge Paragraphs 5 and 6 into the Discussion and retain a concise Conclusion section.
    • Response: We have followed this recommendation by merging the paragraphs into the Discussion section to provide a richer analysis, while the Conclusion section now focuses solely on summarizing the main findings.

Reviewer 2 Report

Comments and Suggestions for Authors

1. Simple summary is missing.

2. Please explain the gender bias in choosing samples.

3. Resolution of images is poor. please amend.

4. Sections such as author contributions, funding, IRB statement and consent are not filled in.

Author Response

  1. Simple Summary Missing:
    • Comment: The "Simple Summary" section was missing in the initial submission.
    • Response: We have now included a "Simple Summary" section, ensuring it complies with the journal's guidelines. This summary provides a clear and non-technical overview of the study, making the content accessible to a broader audience.
  2. Gender Bias in Sample Selection:
    • Comment: Please explain the gender bias in choosing samples.
    • Response: The apparent gender bias in our sample reflects the natural gender distribution of hepatocellular carcinoma (HCC) cases, which is predominantly male. As discussed in the manuscript, this aligns with findings from prior studies, such as Kwon et al. and Ren et al., which observed similar gender distributions in patients treated after liver transplantation. Specifically, Kwon et al. reported 89.5% male patients, while Ren et al. found 90.7% male patients undergoing sequential therapy for advanced HCC. These observations highlight the strong male predominance in HCC due to risk factors such as hepatitis infections and lifestyle-related behaviors, a demographic reality also supported by global cancer statistics (GLOBOCAN) and Turkish Ministry of Health data. This gender distribution is further explored in the Discussion section with reference to relevant literature to provide a comprehensive context.
  3. Resolution of Images:
    • Comment: The resolution of images was deemed insufficient.
    • Response: We have enhanced the resolution of all images in the manuscript, ensuring they meet the recommended standards for publication. The improved images now provide greater clarity and visual quality for readers.
  4. Missing Sections:
    • Comment: Sections such as author contributions, funding, IRB statement, and consent were incomplete.
    • Response: We have now completed these sections as per the journal's requirements:
      • Author Contributions: The roles and contributions of each author have been clearly documented.
      • Funding Statement: We confirm that no external funding was received for this study.
      • IRB Statement: Ethical approval was obtained from Akdeniz University Clinical Research Ethics Committee (protocol #2012-KAEK-20).
      • Informed Consent Statement: Patient consent was waived due to the retrospective nature of the study, as approved by the ethics committee.

Reviewer 3 Report

Comments and Suggestions for Authors

The article entitled 'Use of Sorafenib and Regorafenib in Hepatocellular Cancer Recurrence After Liver Transplantation: Treatment Strategies and Outcomes' by Ozbay et al. highlights the efficacy of sequential Sorafenib and Regorafenib in HCC patients who experienced recurrence after liver transplantation. The article is well written and the data is presented in a clear and concise manner. The following suggestions can be included in the article to improve it. 

1. The article by Kwon et al, (https://www.e-alt.org/journal/view.html?uid=74&vmd=Full) reported a similar study which can be included and discussed. 

2. In simple summary (line 32 to 38) can be removed from the article. 

3. The study include 84.9% male and 15.1% female with 61.5+/-10.9 average age. While comparing to the previously reported progression-free survival (PFS), the age and gender can be included. Example, compare average age of a subjects in a previous study cited. 

Author Response

  1. Inclusion and Discussion of Kwon et al. Study:
    • Comment: The article by Kwon et al. (https://www.e-alt.org/journal/view.html?uid=74&vmd=Full) reported a similar study, which can be included and discussed.
    • Response: We have incorporated a discussion of the study by Kwon et al. in the manuscript. This study provides valuable context regarding demographic distribution and treatment outcomes in HCC patients who experienced recurrence after liver transplantation. The comparison highlights similarities and differences between the findings, enriching the overall discussion.
  2. Simple Summary Refinement:
    • Comment: In the Simple Summary section (lines 32 to 38), the specified lines can be removed.
    • Response: We have refined the Simple Summary section by removing lines 32 to 38 as per your suggestion. This revision ensures the section is more concise and aligned with the core objectives of the study.
  3. Demographics and Comparison with Previous Studies:
    • Comment: The study includes 84.9% male and 15.1% female patients with an average age of 61.5±10.9 years. While comparing to previously reported progression-free survival (PFS), the age and gender can be included. For example, compare the average age of subjects in a previously cited study.
    • Response: We have expanded the demographic analysis in the Results and Discussion sections by comparing the average age and gender distribution of our cohort with those in previous studies, including Kwon et al. and other cited references. For instance, Kwon et al. reported an average age of 60.3 years and a predominantly male cohort (89.5%), which is consistent with our findings. This comparison provides further context and underscores the representativeness of our study population.

Round 2

Reviewer 2 Report

Comments and Suggestions for Authors

We thank the authors for addressing our concerns.